# Validity and Reliability of a Non-Radiographic Postural Analysis Device Based on an RGB-Depth Camera Comparing EOS 3D Imaging: A Prospective Observational Study

**DOI:** 10.3390/healthcare11050686

**Published:** 2023-02-25

**Authors:** Hyo Jeong Lee, Han Eol Cho, Myungsang Kim, Seok Young Chung, Jung Hyun Park

**Affiliations:** 1Department of Rehabilitation Medicine, Bundang Jesaeng Hospital, Seongnam-si 13590, Gyeonggi-do, Republic of Korea; 2Department of Rehabilitation Medicine, Gangnam Severance Hospital, Rehabilitation Institute of Neuromuscular Disease, Yonsei University College of Medicine, Seoul 06229, Republic of Korea; 3Department of Integrative Medicine, Yonsei University College of Medicine, Seoul 06229, Republic of Korea; 4Department of Medical Device Engineering and Management, Yonsei University College of Medicine, Seoul 06229, Republic of Korea

**Keywords:** three-dimensional imaging, posture, skeleton, musculoskeletal pain, full-body X-ray

## Abstract

The posture-analyzing and virtual reconstructing device (PAViR) used a Red Green Blue-Depth camera as a sensor and skeleton reconstruction images were produced. This PAViR quickly analyzed the whole posture from multiple repetitive shots without radiation exposure in clothes and provided a virtual skeleton within seconds. This study aims to evaluate the reliability when shooting repeatedly and to assess the validity compared to parameters of full-body, low-dose X-rays (EOSs) when applied as diagnostic imaging. As a prospective and observational study, 100 patients with musculoskeletal pain underwent an EOS to obtain whole body coronal and sagittal images. The outcome measures were human posture parameters, which were divided by the standing plane in both EOSs and PAViRs as follows: (1) a coronal view (asymmetric clavicle height, pelvic oblique, bilateral Q angles of the knee, and center of seventh cervical vertebra-central sacral line (C7-CSL)) and (2) a sagittal view (forward head posture). A validation of the PAViR compared to the EOSs revealed that C7-CSL showed a moderate positive correlation with that of the EOS (r = 0.42, *p* < 0.01). The forward head posture (r = 0.39, *p* < 0.01), asymmetric clavicle height (r = 0.37, *p* < 0.01), and pelvic oblique (r = 0.32, *p* < 0.01) compared to those of the EOS had slightly positive correlations. The PAViR has excellent intra-rater reliability in people with somatic dysfunction. Except for both Q angles, the PAViR has fair-to-moderate validation when compared to EOS diagnostic imaging in the parameter representing coronal and sagittal imbalance. Although the PAViR system is not yet available in the medical field, it has the potential to become a radiation-free, accessible, and cost-effective postural analysis diagnostic tool after the EOS era.

## 1. Introduction

Posture is defined as the alignment or orientation of the body in an upright position [1]. Posture is related to the muscle length of the muscle activation rather than force [2]. Clinically, posture is evaluated through the ideal gravitational line or vertical line that constitutes anatomical landmarks of the anterior, posterior, and lateral sides of the body [3]. “Good” posture is considered to be a symmetrical alignment, upright, as well as effective at conserving energy [4]; it is important to reduce the risk of injury, certain prolonged static and awkward postures [5], and various cumulative traumatic disorders [6].

Somatic dysfunction is a group of diseases of the musculoskeletal system and connective tissues, defined as an impaired or altered function of the related components of the somatic (body framework) system [7,8,9]. Somatic dysfunction is aggravated by poor posture or results in an abnormal posture that leads to dysfunctional mechanics. It is the target of treatment in manual medicine [10], both chiropractic and osteopathic medicine, and has been classified as International Classification of Diseases 10th Revision, Clinical Modification (ICD-10-CM) Diagnosis Code M99.0 since 2016. Somatic dysfunction is usually assessed via palpatory investigation to evaluate four features: asymmetry (A), range of motion changes (R), tenderness (T), and tissue texture changes (T) [8]. Among them, the evaluation of asymmetry is mainly made up of subjective inspections. If an objective evaluation can be made using quantitative equipment, a more accurate diagnosis will be possible. In addition, if the entire framework were to be imaged rather than a part of the body, it would be more helpful, biomechanically, to understand a patient’s illness. 

Several studies have highlighted the importance of quantifying posture with radiographic or non-radiographic methods [11]. Patients with musculoskeletal pain have a strong desire for diagnosis via imaging tests [12]. Although radiographs using X-rays are the current gold standard protocol for diagnostic images, exposure to ionizing radiation may induce cancer [13]. Therefore, these methods have limited use in sensitive populations, such as adolescents or pregnant women. Non-radiographic methods are available to monitor patient progress without repetitive radiation exposure [14]. Biophotogrammetry, infra-red motion analysis, plumbline, spinal mouse, surface topography, and three-point ultrasound methods have been presented as viable alternatives; however, these methods are directly dependent on both mathematical methods and collection procedures, and few studies have systematically evaluated these methods [14,15].

The EOS imaging system (EOS imaging, Paris, France) is a full-body, low-dose X-ray (EOS) that is used on patients in a weight-bearing position and is based on the X-ray detection technology of Charpak’s chamber, which was awarded the 1992 Nobel Prize in Physics [16,17,18]. Among the current radiographic methods, the EOS has relatively high accuracy. The dose of an EOS single micro-dose X-ray, 2.6 μSv, is much lower than the daily dose of natural background radiation [19], the global average of which is about 2.4 mSv annually from natural sources of radiation [20,21]. Therefore, EOS has previously been used to diagnose abnormalities and malpositions of the spine, pelvis, and lower extremities using numerous parameters [16]. However, high maintenance and labor costs make it less accessible for clinical applications [16]. Additionally, patients are asked to assume an unnatural posture with both hands on the shoulder or zygomatic bones in a confined space that is usually present with the EOS. All of the previously proposed diagnostic imaging approaches, including the EOS, need one person to measure or derive parameters.

Technologies using real-time, three-dimensional (3D) depth cameras have emerged with recent developments in camera and image processing technology. It is possible to repeatedly evaluate and obtain indices for body shape in real time and reconstruct the shape of the spine and skeleton with a specially designed human pose estimation algorithm [22,23,24]. These processes can be performed with no radiation exposure, less space for equipment installation, and relatively lower costs. Additionally, it automatically shows the results without measuring by a physician.

In this study, we propose a new device to evaluate human posture using a Red Green Blue-Depth (RGB-D) camera and an algorithm to reconstruct the virtual skeleton. This posture analyzing and virtual reconstructing device (PAViR) allows for a quick analysis of the person’s whole posture from multiple repetitive shots without radiation exposure and provides a virtual skeleton within seconds. 

Therefore, the research sought to answer the following questions:In people with somatic dysfunction, is PAViR reliable when shooting repeatedly?When applied as diagnostic imaging, is PAViR valid compared to the parameters of EOS?

## 2. Materials and Methods

### 2.1. Participants

Between January 2020 and June 2020, a prospective study was conducted on patients with somatic dysfunction who had been diagnosed with ICD-10 at the department of rehabilitation medicine in a tertiary hospital. Exclusion criteria included patients who were <19 years old; had a body mass index (BMI) > 35 kg/m^2^; had a history of metallic fixation devices after any spine surgery; or who were pregnant or could potentially be pregnant. Demographic data were shown in Table 1. A total of 100 patients (44 males and 56 females) with a mean age of 47.2 years were included in the study. The mean BMI was 23.1 ± 3.5 kg/m^2^. Informed consent for the publication of identifying information/images in an online open access publication was obtained from all subjects. This study was approved by the Institutional Review Board for Clinical Studies at our institution (3-2019-0305). This research was performed in accordance with the relevant guidelines and regulations. In addition, the study was certainly carried out in accordance with the Declaration of Helsinki.

### 2.2. EOS

The full-body, low-dose EOS was performed on each patient. Participants were instructed to stand in the functionally loaded standing position (shoulders flexed to 45° and hands resting on the zygomatic bones) [25,26,27,28] and to stare forward. Standing positions were carefully monitored by a skilled operator who ensured that spinal segments did not compensate for movement while adapting the arms in a confined place. Whole-body coronal and sagittal images were obtained. 

### 2.3. PAViR

The PAViR (Moti Physio, MG solutions, Seoul, Republic of Korea) assessment was performed on each participant in clothes and was performed twice within 2 min by an experienced physician to determine the intra-rater reliability. When commanded by the PAViR system, a participant stands at the front, side, and back as if taking a picture. The physician only needs to ensure that the subject is standing in the position indicated by the laser.

#### 2.3.1. RGB-D Camera

Moti Physio systems use a real-time 3D RGB-D camera (Astra Pro, Orbbec 3D Technology International, Inc. Troy, MI, USA) as a sensor [29,30]. To capture human motion, markerless approaches, convenient and accessible, typically use an RGB-D camera [29]. The PAViR hardware system consists of a display unit, an input unit, an operation unit, and a positioning unit. The display unit serves to visually inform whether the subject is in the correct posture during measurement and to display the final result. The RGB-D camera of the input unit receives the data, and the operation unit processes the data received from the input unit to calculate the image to be displayed on the display unit. Finally, the positioning unit consists of a laser indicator and a floor mat, which illuminates a cross-laser line on the floor at a specified distance from the PAViR and fixes the position of the floor mat relative to the line. Thus, a specific distance and standard position are maintained for the real-time 3D RGB-D camera.

#### 2.3.2. Support Vector Machine

The RGB-D camera captures data in the front, side, and back while the subject stands in a comfortable position for 2–3 s without moving. The system produces a human silhouette from the depth frame data of the camera using the background subtraction method [11,31]. Subsequently, as a processing algorithm, Simple Linear Iterative Clustering performs superpixel segmentation, and the parts are identified using the Support Vector Machine (SVM, Figure 1) [22,23,30,32]. 

#### 2.3.3. Geometric Method

The coordinates of the user depth data are (x, y, z) in mm (Figure 2A). It is necessary to adjust for sensitive clothing or hair and relatively unconcerned skeletal bones. To overcome this problem, we make some assumptions and use the geometric method. This operation is calculated every frame for 2–3 s, and the 28 points are used as the final calculated value based on the last average body part bone point (3D skeleton) and applied to the 3D model (Figure 2). The results derived from this algorithm are presented as an interactive virtual 3D model via a Liquid Crystal Display (LCD) screen, and the coronal and sagittal images are presented as print versions and email-based Portable Network Graphic (PNG) files.

### 2.4. Outcome Measures

The primary outcome was human posture parameters, which were divided by the standing plane in both the EOS and PAViR as follows: (1) a coronal view (asymmetric clavicle height, the pelvic oblique, bilateral Q angles of the knee, and center of 7th cervical vertebra-central sacral line (C7-CSL)) and (2) a sagittal view (forward head posture). In the EOS, a physician measured the above values directly. Six outcomes were measured on each image: ① the angle between the horizontal line and a line connecting the highest clavicle bones, ② the horizontal distance between the midpoints of bilateral iliac crests, ③ and ④ the angle formed between the patella tendon and anterior superior iliac spine, ⑤ the vertical distance between the center of C7 and central sacral line, and ⑥ the angle between the center of C7 and audial canal in Figure 3. 

The unit of the asymmetric clavicle height, bilateral Q angles, and forward head posture is degrees, and the units of the pelvic oblique and C7-CSL are millimeters. However, with the PAViR, the device showed data in degrees immediately after shooting. Secondary outcomes included a data comparison between the PAViR and EOS for validation.

### 2.5. Data Analysis

All statistical analyses were performed using SPSS for Windows, Version 25 (IBM Corp., Armonk, NY, USA). Intraclass correlation coefficients (ICCs) and their 95% confidence intervals were used to determine the intra-rater reliability for PAViR. ICC values > 0.75 represent excellent reliability, values between 0.4 and 0.75 represent fair-to-good reliability, and values < 0.4 represent poor reliability [33]. The relationships between the PAViR measurements and EOS measurements were compared via a paired t-test and correlation analysis (Pearson correlation coefficient). Pearson correlation coefficients and the ICC were characterized as poor (0.00 to 0.20), fair (0.21 to 0.40), moderate (0.41 to 0.60), good (0.61 to 0.80), or excellent (0.81 to 1.00) [34]. The level of significance was set at <0.05 for all statistical tests.

## 3. Results

### 3.1. Outcomes of Measuring with EOS and PAViR

The descriptive outcomes of the coronal and sagittal parameters measured are shown in Table 2. Negative values mean that the left is raised or that the posture is tilted to the left. There is a significant difference between the two devices in pelvic oblique, bilateral Q angle, and C7-CSL. In the EOS, most participants had a position with their torso tilted to the left and head tilted forward. For the Q angle, only one patient had a negative value on the right knee.

### 3.2. Intra-Rater Reliability of PAViR 

All intra-rater correlation coefficients for the coronal (asymmetric clavicle height, pelvic oblique, bilateral Q angle of the knee, C7-CSL) and sagittal view (forward head posture) parameters were > 0.69, and the highest parameter of the PAViR was the C7-CSL (ICC= 0.84) (Table 3).

### 3.3. Validation of PAViR Compared to Parameters of EOS

Primary outcomes were compared to validate the PAViR for each parameter. An analysis adjusted for age, height, weight, and BMI was calculated. Of the PAViR parameters, C7-CSL showed a moderate positive correlation (r = 0.42, *p* < 0.001) with that of the EOS. Forward head posture (r = 0.39, *p* < 0.002), asymmetric clavicle height (r = 0.37, *p* < 0.002), and pelvic oblique (r = 0.32, *p* < 0.002), compared to the results with the EOS, were fair positive correlations, as shown in Table 4. However, there was no significant correlation for the bilateral Q angles of the knee between PAViR and EOS.

## 4. Discussion

To serve as a pilot study, among the asymmetry that is evaluated to diagnose somatic dysfunction, the PAViR values in the coronal (asymmetric clavicle height, the pelvic oblique, Q angles of the knee, C7-CSL) and sagittal views (forward head posture) were compared with the values obtained using the EOS. Intra-rater reliability was good-to-excellent in the newly developed PAViR system. In PAViR values compared to the EOS, all measured values except the Q angle showed fair-to-moderate correlation. Although we adjusted for sensitive clothing or hair and relatively unconcerned skeletal bones via the geometric method, the C7-CSL value, which is simple to capture the midline for the body frame, is thought to be the most consistent, rather than the Q angle with various knee creases. To clarify the validity, further study will be necessary to and should undertake analyses with different clothing or undressed participants. Additionally, there is a need for technological development.

Somatic dysfunction is the impaired or altered function of related components of the somatic (body framework) system [9]. Dysfunction is not defined by localization but by the result of the interplay of a whole chain of different structures. Clinically, patients may complain of pain after an awkward movement, prolonged posture, or overuse of muscles. Somatic dysfunction is aggravated by poor posture or results in an abnormal posture that leads to dysfunctional mechanics [35]. Among features of somatic dysfunction, the evaluation of asymmetry is mainly made up of subjective inspections. If an objective evaluation can be made using quantitative equipment, a more accurate diagnosis will be possible. To evaluate the correlation of its links, it would be more helpful to understand a patient’s condition if the whole body were imaged rather than a part of the body. Therefore, the EOS system is an outstanding equipment for measuring imbalances within current medical field [17,36]. However, the EOS has also several limitations [16,19]

X-rays and EOS are highly accurate and essential for initial evaluation but are not suitable for routine continuous evaluations every few days or weeks due to radiation overexposure or cost concerns. In whole spine X-rays, the amount of radiation can exceed the annual amount of natural background radiation by 2.4 mSv annually. Therefore, given the risk–benefit analysis, it is difficult to recommend a whole spine X-ray to determine the effectiveness of treatment rather than for initial diagnostic purposes. In contrast, an effective dose of an EOS single micro-dose X-ray (2.6 μSv) is less than the amount from one day of natural background radiation [19]. Nonetheless, the EOS is very expensive and can be difficult to access and maintain. Considering the cost-benefit ratio, the EOS may also not be suitable as a routine assessment tool for the follow-up of treatment effectiveness. However, with the PAViR, patients are not exposed to radiation, and the tests can be performed repeatedly without the risk of radiation, even in growing adolescents and young adults of a childbearing age. Since it is extremely cheaper, indeed, of a 100 times difference, there is no burden of regular multiple patient evaluations. Additionally, the installation and test area is less than 3 m^2^; therefore, there is almost no space limitation, and since it is possible to use without disrobing, there is no need for a separate changing space. Since the device automatically derives the measurements without supervision, the physiotherapist can immediately apply it to treatment without going through a doctor. Therefore, the PAViR can be used not only for medical purposes such as manual, chiropractic, and osteopathic therapy, but also for post-exercise performance analysis such as Pilates, yoga, and general workouts.

The repeatability of thew measurement results is critical when using a simple evaluation tool such as the PAViR. The lack of repeatability would make it difficult to evaluate the effectiveness of manual, chiropractic, osteopathic, or exercise therapy. Other non-radiographic instruments, such as biophotogrammetry, infra-red motion analysis, plumbline, spinal mouse, surface topography, and free point ultrasound, rely on both mathematical methods and collection procedures [14,15]. The PAViR intra-rater reliability results in this study were good-to-excellent for all parameters [37].

Computed tomography (CT) scans and biplanar X-ray 3D reconstructions can be measured relatively accurately, but a reconstruction time of 10 min or more is required, and a skilled person is required for reconstruction. In addition, the risk of radiation exposure cannot be excluded [38,39]. Optical methods such as Moiré–Fringe topography, structured light techniques, the Integrated Shape Imaging System, Quantec system, and Orteliuss scanner can be used to detect spinal deformities [40,41,42]. Some clinics have applied these tools to monitor scoliosis, but they are currently difficult to obtain due to the complicated manipulation of the equipment and the inconvenience of patients having to completely disrobe. Since these devices have been designed primarily to evaluate spinal deformities such as scoliosis, only the back view is taken; therefore, shoulder height asymmetry, pelvic obliques, Q angles of the knee, and forward head posture cannot be obtained. Conversely, the PAViR uses a 3D depth camera to collect body surface data, calibrates using human pose estimation (HPE), defines key specific points, and then reproduces the shape of the spine and skeletal system. The subjects can undergo the test without disrobing, and the results are received on screen or via e-mail within 1 min. It can be used conveniently without human effort like the previously mentioned technologies since all processes are automated.

Therefore, the PAViR system could grant the potential to assess the skeletal posture of clothed participants without radiation exposure. Unlike conventional non-radiography, it is a technology that can analyze the whole body in real-time without taking off clothes via the markerless approach of the RGB-D camera. In addition, participants can stand in front of the device and follow the guide without a measurer, and it might be easily transmitted to the medical staff for an application. Further studies are needed to test whether our findings apply to patients with other musculoskeletal disorders, such as cobb angle of scoliosis, degenerative lumbar kyphosis, abnormal pelvic rotation, bowlegs, and knock knees.

### Study Limitations

This study has some limitations. First, although power analysis revealed the statistical significance of our data; assuming a power of 80%, 29 curves were required for each subgroup according to the sample size calculator for the G-power 3.1.9.4 program; including greater numbers of subjects could further increase the power of our study. Second, the algorithm has not been perfected thus far; it is necessary to constantly update the algorithm through comparison with EOS and big data technology. Accumulated data will be helpful to increase the accuracy and correlation by using big data, thus increasing the prediction of risk for musculoskeletal disorders. Third, different positions may result in a variation in the value itself. Although the observed trend would be maintained, more research would be required to compare them in the same position in order to improve accuracy.

## 5. Conclusions

The PAViR has excellent intra-rater reliability in people with somatic dysfunction. Except for both Q angles, the PAViR has fair to moderate validation when compared to EOS diagnostic imaging in the parameter representing coronal and sagittal imbalance. Although the PAViR system is not yet available in the medical field, it has the potential to become a radiation-free, accessible, and cost-effective postural analysis diagnostic tool after the EOS era. This study provides valuable insights to researchers interested in digital healthcare. Based on this technology, more research into improving algorithm accuracy using big data is required, and it will be necessary to apply to patients with musculoskeletal pain.

## Figures and Tables

**Figure 1 healthcare-11-00686-f001:**
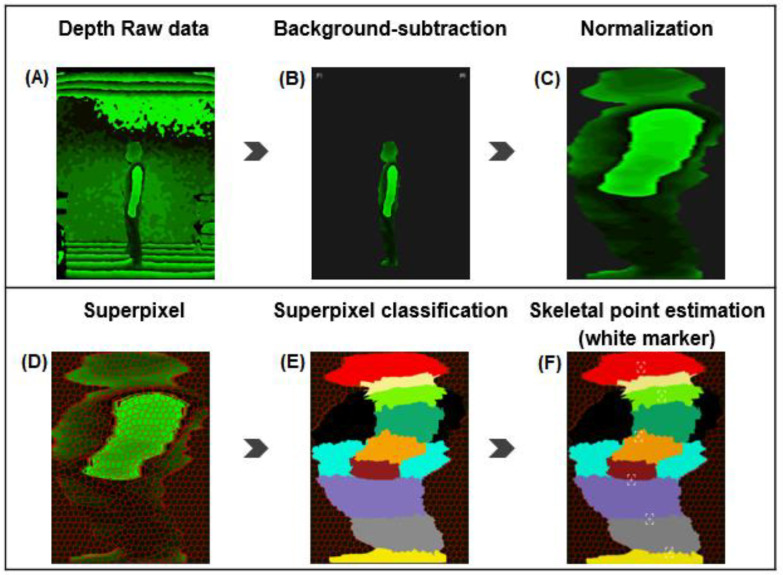
An example of the measurement in the side (**A**). The flow of the human skeletal pose estimation method from the depth camera. After producing a human silhouette using the background-subtraction method (**B**), the human finally becomes normalized (**C**). The classification of body parts with various colors shows an already trained processing algorithm for segmentation via Simple Linear Iterative Clustering (**D**,**E**). Results of the skeletal point estimation on a human body are extracted by using an image processing algorithm. In a markerless object (**E**), white markers are created via a series of processes (**F**). Illustration provided by MG solutions.

**Figure 2 healthcare-11-00686-f002:**
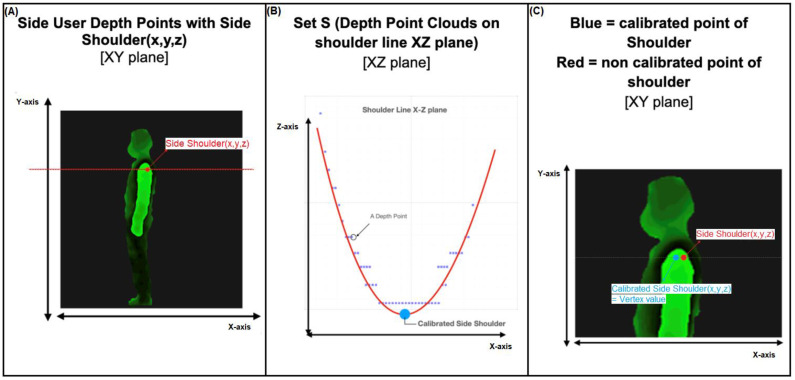
An example of a ‘calibrated skeletal point of side shoulder’ from the side. (**A**) Side user depth points with side shoulder (x, y, z) in the XY plane. (**B**) The set ‘S’ is defined as the depth point clouds on the shoulder line in the XZ plane; these points form a graph of quadratic equations. (**C**) The calibrated point of the shoulder was generated in the XY plane. Illustration provided by MG solutions.

**Figure 3 healthcare-11-00686-f003:**
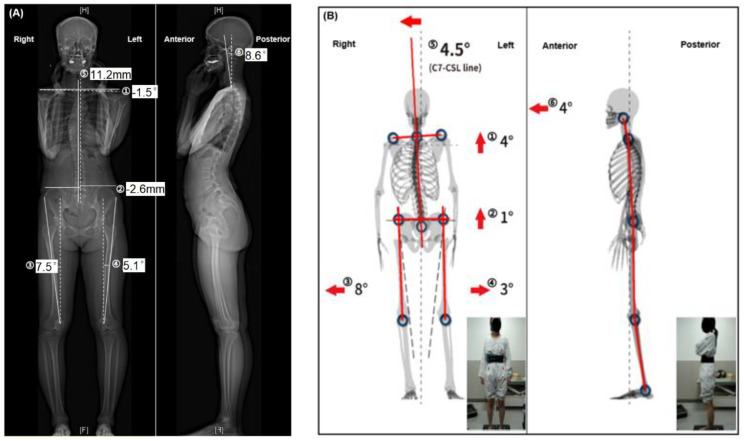
(**A**) The EOS and (**B**) PAViR data in the same subject: ① Asymmetric clavicle height, ② pelvic oblique, ③ right Q angles of the knee, ④ left Q angles of the knee, ⑤ the center of 7th cervical vertebra-central sacral line (C7-CSL), ⑥ forward head posture. All parameters are in degrees except when representing ② and ⑤ in EOS, indicated instead as distance, millimeter. Negative values mean that the left is raised or that the posture is tilted to the left. Illustration provided by MG solutions. PAViR posture analyzing and virtual reconstructing device.

**Table 1 healthcare-11-00686-t001:** Demographic characteristics of participants.

Variables (*n* = 100)	Values	Range
Gender, n male/female	44/56	
Age (y), (mean ± SD)	47.2 ± 16.5	19~81
Weight (kg), (mean ± SD)	63.3 ± 13.1	37.4~89.3
Body height (cm), (mean ± SD)	163.4 ± 19.3	143.0~183.0
Body mass index (kg/m^2^), (mean ± SD)	23.1 ± 3.5	15.6~30.8

SD standard deviation.

**Table 2 healthcare-11-00686-t002:** Descriptive statistics of coronal and sagittal parameters obtained via the EOS and PAViR.

		EOS		PAViR	
	Parameters	Mean ± SD	Range	Mean ± SD	Range
Coronal view	Asymmetric clavicle height (°)	0.1 ± 2.8	−8.0 ^a^~16.0	−1.0 ± 1.6	−5.0~3.8
	Pelvic oblique (mm, °) ^b^	−0.3 ± 5.0	−12.0~14.0	0.7 ± 1.6 *	−2.6~6.4
	Right Q angle (°)	6.1± 1.7	−1.8~10.4	0.9 ± 7.9 *	−8.0~14.0
	Left Q angle (°)	5.6 ± 1.6	0.9~9.8	−3.1 ± 4.0 *	−7.9~13.5
	C7-CSL (mm, °) ^b^	−3.0 ± 13.3	−59.0~36.0	−1.3 ± 2.2 *	−8.1~4.3
Sagittal view	Forward head posture (°)	7.0 ± 6.9	−5.1~29.4	7.9 ± 6.3	−5.0~29.0

PAViR posture-analyzing and virtual reconstructing device, SD standard deviation, C7-CSL center of 7th cervical vertebra-central sacral line; ^a^ Negative values mean that the left is raised or that the posture is tilted to the left; ^b^ All parameters are in degrees except for the pelvic oblique and C7-CSL in the EOS, indicated as distance, millimeter; * *p* < 0.05.

**Table 3 healthcare-11-00686-t003:** Intra-rater reliability of PAViR.

Parameters	Coefficient Value	*p*-Value
Asymmetric clavicle height	0.69	0.005
Pelvic oblique	0.72	0.002
Right Q angle of knee	0.72	0.001
Left Q angle of knee	0.79	0.001
C7-CSL	0.84	0.002
Forward head posture	0.76	0.001

PAViR posture-analyzing and virtual reconstructing device, C7-CSL center of 7th cervical vertebra-central sacral line.

**Table 4 healthcare-11-00686-t004:** The Pearson correlation coefficient (r) for validity between PAViR and EOS.

Parameters	Correlation Coefficient	*p*-Value
Asymmetric clavicle height	0.37	<0.002
Pelvic oblique	0.32	<0.002
Right Q angle of knee	−0.47	0.14
Left Q angle of knee	−0.15	0.15
C7-CSL	0.42	<0.001
Forward head posture	0.39	<0.002

PAViR posture analyzing and virtual reconstructing device, C7-CSL center of 7th cervical vertebra-central sacral line.

## Data Availability

All data analyzed in this study are available from the corresponding author upon reasonable request.

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
