# Peer review of "Validity and Reliability of a Non-Radiographic Postural Analysis Device Based on an RGB-Depth Camera Comparing EOS 3D Imaging: A Prospective Observational Study"

_healthcare, 2023, doi:10.3390/healthcare11050686_

Round 1
Reviewer 1 Report
ABSTRACT
The conclusion in the abstract is more a presentation of the device than the conclusions of the article. The conclusion must respond to the objective of the study and, as it is not responding, I suggest changing the conclusion of the abstract.
INTRODUCTION
- The authors do not make it clear what would be the difference between this device and many others available in the literature, which assess posture by means of photogrammetry. I suggest doing a comparison of the devices.
METHODS
- Why did the authors use this posture: (shoulders flexed to 45° and hands resting on the zygomatic bones)? This needs to be clear in the manuscript. What is the reference for this measurement?
- This type and posture could not have influenced the results? Why did the authors not collect data with arms extended along the body? All this has to be described in the text.
- The study has a very large sample in relation to age, could this not have influenced the results? Why not perform an analysis by age group?
- Why didn't the volunteers perform the study in bathing suits?
- Was the PAVIR assessment performed by the same physician twice? You need to make this clear in the text.
- Item 2.3.2 is very badly described, in fact, it is incomprehensible. When authors intend to publish in a multidisciplinary journal, they have to use easier language, even scientific. As it stands, it is confusing and incomprehensible how the authors performed these measurements.
- Outcome evaluation: the authors mention that the doctors measured it, but they do not explain how the measurement was carried out, making it difficult to understand the evaluation of the outcomes.
- The English throughout the entire manuscript must be improved.
- The statistical analysis should be better organized, I suggest that the authors explain the analyzes according to the outcomes. In addition, they should better organize the sequence of the analysis, for example, would the t-test be used before comparing which groups? Before this test we have to use the test to analyze the sample distribution, not mentioned by the authors. Statistical analysis needs to be rewritten.
RESULTS
- Demonstrate in the tables which tests were used, with some symbol.
- In addition, it is correct to put the exact value of the p-value and not expressions such as p<0.001, the p-values must come from the table itself.
- Where did the authors use the t-test?
DISCUSSION
- The discussion is vague and does not focus on the validation of this instrument, comparing it with other equivalent instruments for postural assessment, demonstrating the strengths and weaknesses of each one of them, their benefits and costs.
- Discussion is the poorest part of the study.
- The uncalculated sample size should be a limitation of the study, since the sample was not estimated through sample calculation.
CONCLUSION
- The conclusion, as in the abstract, is more a recommendation of the device, than, in fact, the conclusions of the article. The conclusion must respond to the objective of the study and, as it is not responding, I suggest changing the conclusion. Any other recommendations should come at the end of the discussion.
Author Response
For the reviewer 1
ABSTRACT
The conclusion in the abstract is more a presentation of the device than the conclusions of the article. The conclusion must respond to the objective of the study and, as it is not responding, I suggest changing the conclusion of the abstract.
Response: Thank you for your comments. I agree with your opinion. To clarify, we changed the conclusion of the abstract. “The PAViR has excellent intra-rater reliability in people with somatic dysfunction. Except for both Q-angles, the PAViR has fair to moderate validation when compared to EOS diagnostic imaging in the parameter representing coronal and sagittal imbalance. Although the PAViR system is not yet available in the medical field, it has the potential to become a radiation-free, accessible, and cost-effective postural analysis diagnostic tool after the EOS era,” we added.
INTRODUCTION
- The authors do not make it clear what would be the difference between this device and many others available in the literature, which assess posture by means of photogrammetry. I suggest doing a comparison of the devices.
Response: Thank you for your feedback. We revised the sentence, “Biophotogrammetry, infra-red motion analysis, plumbline, spinal mouse, surface topography, and free point ultrasound methods have been presented as viable alternatives; however, these methods are directly dependent on both mathematical methods and collection procedures, and few studies have systematically evaluated these methods,” and added a reference1 that included photogrammetry as a tool for postural evaluation as a systemic review.
- Furlanetto TS, Sedrez JA, Candotti CT, Loss JF. Photogrammetry as a tool for the postural evaluation of the spine: A systematic review. World J Orthop. 2016 Feb 18;7(2):136-48.
METHODS
- Why did the authors use this posture: (shoulders flexed to 45° and hands resting on the zygomatic bones)? This needs to be clear in the manuscript. What is the reference for this measurement?
Response: Thank you for your comments. We followed the standing position of previous studies for EOS. The EOS had been already proven to have ability in reconstructing 3D images in the functional standing position. The functional standing position means that it covers the evaluation of sagittal alignment and compensatory mechanisms to restore its asymmetry. Many studies applied to position in the functionally loaded standing position: the participant was standing naturally, gaze forward, elbows fully flexed with fingers resting on the cheek bones or zygomaticus.1-3 Therefore, the position was fixed to protocol in EOS. To improve comprehension, we changed “modified standing position” to “functionally loaded standing position” and added several references.
- Garg B, Mehta N, Bansal T, Malhotra R. EOS® imaging: Concept and current applications in spinal disorders. J Clin Orthop Trauma. 2020 Sep-Oct;11(5):786-793.
- Pumberger M, Schmidt H, Putzier M. Spinal Deformity Surgery: A Critical Review of Alignment and Balance. Asian Spine J. 2018 Aug;12(4):775-783.
- Vergari C, Skalli W, Clavel L, Demuynck M, Valentin R, Sandoz B, Similowski T, Attali V. Functional analysis of the human rib cage over the vital capacity range in standing position using biplanar X-ray imaging. Comput Biol Med. 2022 May;144:105343.
- This type and posture could not have influenced the results? Why did the authors not collect data with arms extended along the body? All this has to be described in the text.
Response: Thank you for your comments. The EOS had already established a protocolized posture, and the PAViR was looking for a posture to minimize variables in extracting values. Although the value itself may be different, it was judged that the trend of the observed outcome would be maintained. This was explained in the study limitations. “Third, different positions may result in a variation in the value itself. Although the observed trend would be maintained, more research would be required to compare them in the same position in order to improve accuracy,”
- The study has a very large sample in relation to age, could this not have influenced the results? Why not perform an analysis by age group?
Response: Thank you for your comments. In the method of extracting values ​​through the PAViR system, there was no factor affected by age. Therefore, an analysis by age group was not performed.
- Why didn't the volunteers perform the study in bathing suits?
Response: Thank you for your comments. Since the EOS used radiography and the PAViR analyzed skeletal structure through a depth camera while clothed, we designed this method including calibrated technology. However, as you mentioned, a way to measure undressed person’s posture parameters seems to be a good study to increase the accuracy and validity of PAViR system. The content was included in the Discussion. “To clarify the validity, further study will be necessary to be investigated with analyses with different clothing or undressed participants”
- Was the PAVIR assessment performed by the same physician twice? You need to make this clear in the text.
Response: Thank you for your comments. “By an experienced physician to determine the intra-rater reliability,” we explained.
- Item 2.3.2 is very badly described, in fact, it is incomprehensible. When authors intend to publish in a multidisciplinary journal, they have to use easier language, even scientific. As it stands, it is confusing and incomprehensible how the authors performed these measurements.
Response: Thank you for your comments. I agree with your opinion. It appears to lean toward machine description and technology including formula. To clarify, we revised and removed confusing content from the 2.3. PAViR description. Please refer to the method’s modifications.
- Outcome evaluation: the authors mention that the doctors measured it, but they do not explain how the measurement was carried out, making it difficult to understand the evaluation of the outcomes.
Response: Thank you for your comments. When commanded by the PAViR system, a participant stands the front, side, and back as if taking a picture. The physician only needs to ensure that the subject is standing in the position indicated by the laser, which is completed within two minutes. This content was added in the Materials and Methods. “When commanded by the PAViR system, a participant stands the front, side, and back as if taking a picture. The physician only needs to ensure that the subject is standing in the position indicated by the laser”
- The English throughout the entire manuscript must be improved.
Response: Thank you for your comments. We additionally corrected the paper and attached the English editing certificate.
- The statistical analysis should be better organized, I suggest that the authors explain the analyzes according to the outcomes. In addition, they should better organize the sequence of the analysis, for example, would the t-test be used before comparing which groups? Before this test we have to use the test to analyze the sample distribution, not mentioned by the authors. Statistical analysis needs to be rewritten.
Response: Thank you for your comments. I agree with your opinion. We added the result of t-test and revised vague contents in 3.1. Outcomes of measuring with EOS and PAViR.
Added text - “There is a significant difference between two devices in Pelvic oblique, bilateral Q-angle, and C7-CSL”
Deleted text – “The outcomes of C7-CSL and forward head posture also showed a similar relationship to measurements of PAViR”
Revised text – “The partial correlation coefficient” to “The Pearson correlation coefficient” in Table 4.
RESULTS
- Demonstrate in the tables which tests were used, with some symbol.
Response: Thank you for your comments. I agree with your opinion. We added the result of t-test with a symbol in Table 2.
- In addition, it is correct to put the exact value of the p-value and not expressions such as p<0.001, the p-values must come from the table itself.
Response: Thank you for your comments. We revised the Table 3, 4 with the p-values.
- Where did the authors use the t-test?
Response: Thank you for your comments. I agree with your opinion. We added the result of t-test in 3.1. Outcomes of measuring with EOS and PAViR.
DISCUSSION
- The discussion is vague and does not focus on the validation of this instrument, comparing it with other equivalent instruments for postural assessment, demonstrating the strengths and weaknesses of each one of them, their benefits and costs.
Response: We appreciate your comments. I agree with your opinion. We generally changed and revised in discussion.
- Discussion is the poorest part of the study.
Response: We appreciate your comments. We generally changed and revised in discussion.
- The uncalculated sample size should be a limitation of the study, since the sample was not estimated through sample calculation.
Response: Thank you for your comments. I agree with your opinion. We expressed the uncalculated sample size for G-power in study limitations.
CONCLUSION
- The conclusion, as in the abstract, is more a recommendation of the device, than, in fact, the conclusions of the article. The conclusion must respond to the objective of the study and, as it is not responding, I suggest changing the conclusion. Any other recommendations should come at the end of the discussion.
Response: Thank you for your comments. We revised the conclusion in response to your feedback.

Reviewer 2 Report
The paper presents a method for evaluating the reliability of repeated photography and validation against whole-body low-dose x-ray (EOS) parameters when a diagnosis is made using imaging.
The manuscript requires some significant modifications.
1. Introduction
- line 46, which is the difference between sensitivity (T) and changes in tissue texture (T), if they are marked with the same letter
- line 65, the x-ray is written with a small letter, and the rest appears written with X
- The contributions and importance of the paper should be highlighted. What are the main advantages of this article concerning other similar works?
- You can also add the article structure. This thing will be helpful for the reader to get an idea of the article layout.
2. Materials and Methods
- Define SD (Table 1, etc.)
-- figures 1 and 2 should be uploaded in a more explicit version so that they can be viewed more easily
- lines 157 and 158 are not formatted correctly; there should be a space
- You marked with S a set of points. Why did you write it down, and why is it helpful to you further in your study, considering that it no longer appears anywhere?
- In line 171, the terms r and f are not defined. Also, number the respective relationship
4. Discussion
- line 287, the x-ray is written with a small letter, and the rest appears written with X
- highlight study limitations with a Study Limitations subtitle
5. Conclusions
- In the introduction, you presented two questions to which you had to find the answer. Therefore, these answers should appear here.
- What are the future works?
- What are the main challenges of the current work?
References
References should be more current. However, you can also find references from the last five years.
Author Response
For the reviewer 2
The paper presents a method for evaluating the reliability of repeated photography and validation against whole-body low-dose x-ray (EOS) parameters when a diagnosis is made using imaging.
The manuscript requires some significant modifications.
- Introduction
- line 46, which is the difference between sensitivity (T) and changes in tissue texture (T), if they are marked with the same letter
Response: Thank you for your comments. Somatic dysfunction is claimed to be diagnosed by palpation using four cardinal clinical features: tenderness, asymmetry, range of motion abnormality, and tissue texture changes.1-4 The mnemonic TART or ARTT is commonly mentioned as a memory aid by various studies. Some authors do not include tenderness as a clinical sign1 or substitute ‘sensitivity’ for tenderness.2 Tenderness is the subjective sensation of pain or soreness in response to palpation of tissues, which means sensitivity alterations. However, several studies described T as suggestive of changes in tissue texture.
- Greenman PE. Principles of manual medicine. 3rd ed. Philadelphia: Lippincott William & Wilkins; 2003.
- DiGiovanna EL, Schiowitz S, Dowling DJ. An osteopathic approach to diagnosis & treatment. 3rd ed. Philadelphia: Lippincott William & Wilkins; 2005.
- Gibbons P, Tehan P. Manipulation of the spine, thorax and pelvis. An osteopathic perspective. 3rd ed. London: Churchill Livingstone; 2008.
- Ehrenfeuchter WC, Kappler RE. Palpatory examination. Foundations of osteopathic medicine. 3rd ed. Philadelphia, PA: Lippincott Williams & Wilkins; 2011. p. 401e9.
- line 65, the x-ray is written with a small letter, and the rest appears written with X
Response: Thank you for your comments. We revised the X-ray in upper case.
- The contributions and importance of the paper should be highlighted. What are the main advantages of this article concerning other similar works?
Response: Thank you for your comments. In the medical profession, EOS is already utilized in 3D analysis based on X-rays that can be conveniently accessible when assessing the anatomy of the musculoskeletal system. However, compared to EOS, the PAViR is fast and easy to use, including for pregnant women and children with an unchanged suits. Because it captures the camera without radiation exposure through a markerless approach and displays the parameters directly on the screen via the hardware system. The content was included in the Discussion. “Therefore, the PAViR system could be the potential to assess the skeletal posture of clothed participants without radiation exposure. Unlike conventional non-radiography, it is a technology that can analyze the whole body in real-time without taking off clothes by markerless approaching RGB-D camera. In addition, participants can stand in front of the device and follow the guide without a measurer, and it might be easily transmitted to the medical staff for an application”
- You can also add the article structure. This thing will be helpful for the reader to get an idea of the article layout.
Response: Thank you for your comments. We attached the article structure to comprehend the reader.
- Materials and Methods
- Define SD (Table 1, etc.)
Response: Thank you for your comments. We described SD, standard deviation.
-- figures 1 and 2 should be uploaded in a more explicit version so that they can be viewed more easily
Response: Thank you for your comments. We changed the figures to improve resolution.
- lines 157 and 158 are not formatted correctly; there should be a space
Response: Thank you for your comments. We had a space and reviewed this script for format.
- You marked with S a set of points. Why did you write it down, and why is it helpful to you further in your study, considering that it no longer appears anywhere?
Response: Thank you for your comments. I agree with your opinion. It appears to lean toward machine description and technology including formula. To clarify, we revised and removed confusing content from the 2.3. PAViR description. To demonstrate the difference of the new device, the calibration method preserved Figure 2.
- In line 171, the terms r and f are not defined. Also, number the respective relationship
Response: Thank you for your comments. We removed the formula to make it easier to understand.
- Discussion
- line 287, the x-ray is written with a small letter, and the rest appears written with X
Response: Thank you for your comments. We revised the X-ray in upper case.
- highlight study limitations with a Study Limitations subtitle
Response: Thank you for your comments. We added the paragraph with a Study limitations.
- Conclusions
- In the introduction, you presented two questions to which you had to find the answer. Therefore, these answers should appear here.
Response: Thank you for your comments. The PAViR has excellent intra-rater reliability in people with somatic dysfunction. Except for both Q-angles, the PAViR has fair to moderate validation when compared to EOS diagnostic imaging in the parameter representing coronal and sagittal imbalance. Although the PAViR system is not yet available in the medical field, it has the potential to become a radiation-free, accessible, and cost-effective postural analysis diagnostic tool after the EOS era.
- What are the future works?
Response: Thank you for your comments. To improve the validity, accumulated data will be used and software in PAViR system has to be updated through big data technology. Also, a method of comparing the result values of the same position will also help improve the algorithm. We presented the contents to supplement in study limitations.
- What are the main challenges of the current work?
Response: Thank you for your comments. It is helpful to increase validity and apply it to disease other than somatic dysfunction. We added the contents at the end of the discussion.
References
References should be more current. However, you can also find references from the last five years.
Response: Thank you for your comments.

Reviewer 3 Report
The authors proposed a device to assess human posture through a Red Green Blue-depth camera. They also showed the implementation of an algorithm to reconstruct a virtual skeleton. Thus, the authors aimed for answering the reliability of PAViR when shooting repeatedly, and its validity compared against EOS parameters.
Although, overall, the paper is well written and documented, I’m having three main concerns as follows:
· 1. The authors claim to have used SVM to identify subject’s body parts, but they neither reference the data used for the training nor the performance of the model generated by the SVM (accuracy, for instance).
· 2. Similarly, the authors argue to have used Pearson’s correlation coefficient and paired t-tests. However, I couldn’t explicitly find the results of any of these tests in the paper. Furthermore, there are some descriptions of Pearson’s correlation coefficient in lines 234 onwards, but this test is inappropriate to assess Validity.
· 3. I don’t see how the discussion an conclusion sections are backed up by the results presented in the paper.
Could the authors clarify about these?
Author Response
For the reviewer 3
The authors proposed a device to assess human posture through a Red Green Blue-depth camera. They also showed the implementation of an algorithm to reconstruct a virtual skeleton. Thus, the authors aimed for answering the reliability of PAViR when shooting repeatedly, and its validity compared against EOS parameters.
Although, overall, the paper is well written and documented, I’m having three main concerns as follows:
- The authors claim to have used SVM to identify subject’s body parts, but they neither reference the data used for the training nor the performance of the model generated by the SVM (accuracy, for instance).
Response: Thank you for your comments. We cited another SCIE-accepted manuscript that was researched on a different topic using the PAViR. In addition, we added a paper that referenced the technology. We don’t mention it because the goal is to compare estimates following SVM classification, and so on. To improve accuracy, we agree that these steps should be confirmed in future studies.
- Similarly, the authors argue to have used Pearson’s correlation coefficient and paired t-tests. However, I couldn’t explicitly find the results of any of these tests in the paper. Furthermore, there are some descriptions of Pearson’s correlation coefficient in lines 234 onwards, but this test is inappropriate to assess Validity.
Response: Thank you for your comments. There must have been some ambiguity in the results analysis. We modified “partial” to “Pearson” in Table 4. We also added the paired t-test result, “There is a significant difference between two devices in Pelvic oblique, bilateral Q-angle, and C7-CSL,” and deleted sentence “The outcomes of C7-CSL and forward head posture also showed a similar relationship to measurements of PAViR.”
It is appropriate to compare to EOS, which is already used in the medical field, and please refer to previous studies1,2 that assessed validity using Person’s analysis.
- Masson, L., MCNeill, G., Tomany, J., Simpson, J., Peace, H., Wei, L., Grubb DA, Bolton-Smith, C. (2003). Statistical approaches for assessing the relative validity of a food-frequency questionnaire: Use of correlation coefficients and the kappa statistic. Public Health Nutrition, 6(3), 313-321.
- Campos S, Zhang L, Sinclair E, Tsao M, Barnes EA, Danjoux C, Sahgal A, Goh P, Culleton S, Mitera G, Chow E. The palliative performance scale: examining its inter-rater reliability in an outpatient palliative radiation oncology clinic. Support Care Cancer. 2009 Jun;17(6):685-90.
- I don’t see how the discussion an conclusion sections are backed up by the results presented in the paper.
Response: Thank you for your comments. We changed the conclusion according to two questions in the introduction. We also generally revised the discussion. Please refer to the discussion.
Could the authors clarify about these?
Response: Thank you for your comments.

Round 2
Reviewer 1 Report
Congratulations to the authors, I believe that after these modifications, the text became clearer to the readers of how the entire construction of the article was. I now believe the article is ready for publication.
Author Response
Thanks for that.
Reviewer 2 Report
Please check the formatting of both the text and the figures. Some of the figures are offset from their title.
Author Response
Thank you for your feedback. We generally checked and revised the formatting of the text and the figures. Please refer to the manuscript.